# Metagenome-Assembled Genome of *Cyanocohniella* sp. LLY from the Cyanosphere of Llayta, an Edible Andean Cyanobacterial Macrocolony

**DOI:** 10.3390/microorganisms10081517

**Published:** 2022-07-27

**Authors:** Claudia Vilo, Qunfeng Dong, Alexandra Galetovic, Benito Gómez-Silva

**Affiliations:** 1Laboratory of Biochemistry, Biomedical Department, Health Sciences Faculty and Centre for Biotechnology and Bioengineering (CeBiB), Universidad de Antofagasta, Antofagasta 1270300, Chile; claudiavilo@gmail.com (C.V.); alexandra.galetovic@uantof.cl (A.G.); 2Center for Biomedical Informatics, Department of Medicine, Stritch School of Medicine, Loyola University of Chicago, Chicago, IL 60660, USA; qdong@luc.edu

**Keywords:** cyanosphere, cyanobacteria, *Cyanocohniella*, Llayta, macrocolonies, metagenomics, metagenomic-assembled genome, microbiome

## Abstract

Cyanobacterial macrocolonies known as Llayta are found in Andean wetlands and have been consumed since pre-Columbian times in South America. Macrocolonies of filamentous cyanobacteria are niches for colonization by other microorganisms. However, the microbiome of edible Llayta has not been explored. Based on a culture-independent approach, we report the presence, identification, and metagenomic genome reconstruction of *Cyanocohniella* sp. LLY associated to Llayta trichomes. The assembled genome of strain LLY is now available for further inquiries and may be instrumental for taxonomic advances concerning this genus. All known members of the *Cyanocohniella* genus have been isolated from salty European habitats. A biogeographic gap for the *Cyanocohniella* genus is partially filled by the existence of strain LLY in Andes Mountains wetlands in South America as a new habitat. This is the first genome available for members of this genus. Genes involved in primary and secondary metabolism are described, providing new insights regarding the putative metabolic capabilities of *Cyanocohniella* sp. LLY.

## 1. Introduction

Cyanobacterial and microalgal biomasses have been consumed for centuries [1,2,3,4,5,6,7]. *Dunaliella*, *Chlorella*, *Arthrospira,* and *Nostoc* are documented sources of essential amino acids [4,5,6], vitamins [5,6,7,8], polyunsaturated fatty acids [5,6,7,9,10], carotenoids [2,11], phycobiliproteins [5,11,12], and secondary metabolites [5,7,8].

The Atacama Desert is considered the driest and oldest desert on Earth, with climatic, oceanic, latitudinal, and geomorphological factors that account for its existence [13,14,15]. Precipitations at the Andes Range are at least 40–50 fold higher than those observed at the dry core of the Atacama Desert [15]. Indigenous cultures have historically acknowledged the Andean biome as source of forage, food, and ethnomedicine [16,17,18]. The consumption of cyanobacterial macrocolonies made of filamentous diazotrophic species, known by the vernacular name of Llayta, is a centenary Andean alimentary practice that can be traced back to pre-Columbian times and it is an example of a neglected natural food resource [10,16,19,20,21,22]. Llayta macrocolonies are collected from wetlands in the Andes highlands over 3000 m above sea level; they are sun-dried and sold for human consumption as a dry, dark green, leaf-like biomass in food markets in Arica and Iquique, in northern Chile, and Tacna, in southern Peru [10,16,19]. Ethnographic, chemical composition, and cyanotoxic studies in addition to the absence of negative epidemiological evidence, support the notion that Llayta would be a safe, natural food ingredient [10,16,23,24,25].

The natural life cycle of primary producers may involve mutualistic, antagonistic, or commensal interactions with cyanophages, fungi, and heterotrophic bacteria, and positive or negative outcomes have been previously discussed in *Nostoc* macrocolonies [26,27,28,29]. However, identification of members of the bacterial community associated with edible Llayta macrocolonies have been a pending issue. Here, we report the metagenome-assembled genome (MAG) of a previously undetected cyanobacterium strain associated to the cyanosphere of isolated Llayta trichomes. We have identified it as a member of the genus *Cyanocohniella*, referred here as strain *Cyanocohniella* sp. LLY, expanding the presence of the genus *Cyanocohniella* to a new habitat, the Andes Mountains wetlands in South America. Bioinformatic analyses of MAG allowed the identification of this new strain, the annotation of functional genes, and insights regarding its putative metabolic capabilities. The reconstructed genome of strain LLY is now available and instrumental for further inquiries and taxonomic advances on the *Cyanocohniella* genus. 

## 2. Materials and Methods

*Isolation of Llayta filaments.* Dried Llayta colonies were obtained from the major farmers’ market in Arica, Chile, during 2015 and maintained in the original plastic bag until used. Dry colonies were suspended in sterile deionized water for rehydration for 24 h. To isolate filaments, aliquots (5 mL) were transferred to liquid, nitrogen-free Arnon mineral medium [30] and cultured at 30 °C under white fluorescent light (μE 180 m^−2^ s^−1^), continuous agitation (200 rpm), and aeration enriched in 1% *v*/*v* CO_2_ [10,19]. Culture aliquots were spread onto agar plates, observed under a stereo microscope, and isolated filaments were picked and transferred to fresh medium for growth as above. Aliquots (10–15 mL) of cultured Llayta filaments were collected at the exponential growth phase (day 21st), washed with fresh medium, the filaments were recovered as a pellet by centrifugation at 4000× *g*, for 5 min, at room temperature, and used to extract total genomic DNA.

*DNA extraction.* Total genomic DNA was extracted from the filament pellets with Ultra Clean Microbial DNA isolation kit (MoBio Labs., Inc., Carlsbad, CA, USA), following the manufacturer’s instructions. DNA quality was evaluated by electrophoresis in 0.8% agarose gel and quantified photometrically at 260 nm.

*DNA sequencing and analysis.* The metagenome of Llayta filaments was sequenced via MiSeq sequencing technology using shotgun paired-end libraries, with an average insert size of 250 bp. Reads had an average length of 300 bp, with good quality scores, as evaluated by the FastQC program (version 0.10.0). The sequencing produced a total of 17,137,246 reads. Metagenomic assembly was done using MEGAHIT assembler v.1.2.9 [31], and binning was conducted using the PATRIC web server [32]. The complete genome was annotated using the Rapid Annotations using Subsystem Technology (RAST) server version 4.0 [33]. Secondary metabolites were searched with PRISM version 4.4.4 [34] and AntiSMASH version 6.0 [35] software. 

*Data Deposition*. Sequencing reads are available at the Sequence Read Archive (SRA) with accession number SRR17916224. The Whole Genome Shotgun project have been deposited at DDBJ/ENA/GenBank under the accession JAKOMP000000000. 

## 3. Results

### 3.1. Genome Assembly and Phylogeny of Cyanocohniella sp. LLY

A preliminary metagenome analysis of Llayta trichomes reads revealed predominant bacteria belonging to the phylum *Proteobacteria* (82%). Assembly and binning strategy resulted in four reconstructed bacterial genomes resembling the bacterial genera *Aquimonas (Gammaproteobacteria), Microvirga (Alphaproteobacteria), Mesorhizobium (Alphaproteobacteria) and Paracoccus (Alphaproteobacteria)*. Most importantly, the metagenome analysis revealed that 16% of the reads corresponded to the phylum Cyanobacteria, from which one cyanobacterium genome was reconstructed by metagenomic binning. The MAG was assigned to the genus *Cyanocohniella* based on 16S-rRNA gene analysis, which is consistent with the dominance of Nostocales reads (40%) among the cyanobacteria phylum. The *Cyanocohniella* sp. LLY genome was reconstructed using 244 contigs with an N50 of 50,369, which rendered a total genome size of 6,781,030 bp with a GC content of 41.2%. The binning strategy accounted for a complete reconstructed genome with minimal contamination (0.7%).

Phylogenetic analyses of *16S rRNA* gene confirmed that the MAG from Llayta filaments belonged to the order Nostocales. Moreover, the 16S rRNA gene analysis revealed a placement of Llayta within a clade of *Cyanocohniella* sequences (Figure 1). A BLASTN search with the Llayta MAG’s 16S rRNA in the NCBI nr database showed a sequence identity of 99.2% with *Cyanocohniella* sp. SY-1-2-Y (NCBI accession number: MT946558.1). 

### 3.2. Functional Capabilities of Cyanocohniella sp. LLY

The reconstructed genome of *Cyanocohniella* sp. LLY was analyzed by The Rapid Annotation using Subsystem Technology (RAST) server for genome annotation based on subsystem annotations. As expected, the cyanobacterium genomic capabilities included sequences related to the production of vitamins, prosthetic groups, and pigments (5.5%); DNA metabolism (5.3%); fatty acids, lipids, and isoprenoids (2.9%); dormancy and sporulation (0.14%); multiple gene copies associated to Photosystems I and II proteins; genes and pathways associated to DNA repair metabolism, and genes for cell division and cell cycle processes. In addition, the *nif* gene cluster for nitrogen fixation was also identified in LLY MAG. Gene sequences involved in chemotaxis and motility were absent. 

In silico genomic analysis demonstrated the presence of conserved lycopene cyclase *CruA/CruP* genes, the complete *mysABCD* and *scyABCDEF* (and accompanied regulator and kinase sensor) gene clusters involved in carotenoids biosynthesis, mycosporine-like amino acids (MAAs), and scytonemin biosynthesis (Table 1). These putative capabilities are intimately related to protection against photooxidative damages triggered by high-UV solar insolation prevailing in the Andes plateau wetlands [36].

Bioinformatic analyses in the reconstructed genome of *Cyanocohniella* sp. LLY rendered two copies of gene *cphA1* coding for a putative non-ribosomal cyanophycin synthetase. Sequence alignment of *Cyanocohniella* sp. LLY cyanophycin synthetase to Nostocales CphA1 (NCBI accession number: MBE9052550.1 and MBE9053887.1) showed an 85% identity. The reconstructed genome of *Cyanocohniella* sp. LLY also contained three copies of the gene *cphB* coding for a putative cyanophycin-degrading cyanophycinase, which showed an 83% identity with Cyanobacteria cyanophycinase proteins (NCBI accession number: AHJ26983.1 and WP_096726859.1).

CRISPR-Cas (Clustered Regularly Interspaced Short Palindromic Repeats-CRISPR-associated) operons are defensive systems found in archaea and bacteria [37,38,39,40,41,42,43] and were found in the *Cyanocohniella* sp. LLY genome. One of them included *Cmr2-Cmr3-Cmr4-Cmr5-Cmr6* and *Cas1-Cas2* operons flanked by CRISPR arrays. The second system showed the *Csc3-Csc2-Csc1-Cas6-Cas4-Cas1-Cas2* genes and CRISPR array, and the third comprised *Cmr2-Cmr3-Cmr4-Cmr5-Cmr6* genes flanked by CRISPR array (Table 2). Therefore, and according to the observed organization, the CRISPR-Cas system operons in *Cyanocohniella* sp. LLY would indeed confer immunity defense against mobile elements.

### 3.3. Secondary Metabolism of Cyanocohniella sp. LLY

Identification of functional genes and gene clusters coding for enzymes involved in secondary metabolism was carry out by AntiSMASH and PRISM softwares. Secondary metabolites analysis rendered candidate genes that were further analyzed to confirm their presence within the *Cyanocohniella* sp. LLY MAG and to analyze phylogeny among the cyanobacterial phylum.

*Cyanocohniella* sp. LLY MAG contained sequences of the gene *LanM,* identified by both software programs, AntiSMASH and PRISM. *LanM* gene is involved in the biosynthesis of a family of ribosomally synthesized and post-translationally modified peptides. This sequence and the CCG motif for Zn binding sites at the C-terminal showed a 70% identity after alignments with the *LanM* sequence of cyanobacteria bacterium UBA11372 (NCBI accession number HAX78725.1). Figure 2 shows the domains for dehydratase and cyclization activities and three cysteine residues associated with zinc-binding sites, one cysteine residue alone, and the CCG conserved motif from the Llayta cyanobacterium.

The reconstructed genome of *Cyanocohniella* sp. LLY contains sequences for all genes coding for lasso peptide biosynthesis: a lasso peptide precursor lasso P (49% identity to Nostocales, cyanobacterium accession number MBE9052603.1), a lasso C isopeptide bond-forming cyclase (74% identity to Nostocales, cyanobacterium MBE9052604.1), a lasso peptide biosynthesis B2 protein (73% identity to Nostocales, cyanobacterium accession number MBE9052605.1), and lasso peptide biosynthesis B1 PqqD family protein (86% identity to Nostocales, cyanobacterium accession number MBE9052606.1). Lasso C was identified by AntiSMASH and PRISM software. However, other lasso peptide biosynthesis genes were manually curated. Organization for the lasso peptide gene cluster in the genome of the Llayta cyanobacterium, resembled that from Actinobacteria. Figure 3A depicts the putative primary structure of the cleaved lasso peptide from *Cyanocohniella* sp. LLY, the location of the lasso structure with a glycine residue at the peptide N-terminal and bulky residues at its C-terminal, and a comparison of conserved sequences among several microbial genera.

The genome of *Cyanocohniella* sp. LLY also has putative biosynthetic pathways for capreomycidine peptides from non-ribosomal peptide synthases (NRPS), which were identified by PRISM software, and microviridin peptides from ribosomally synthesized and post-translationally modified peptides (RiPPs) identified by AntiSMASH software. Further analysis indicated that the capreomycidine synthase gene from *Cyanocohniella* sp. LLY showed an 82% identity with capreomycidine synthase gene (sequence ID: WP_104904844.1) of a *Nostoc* sp. cyanobiont of *Lobaria pulmonaria* and 42.23% identity with capreomycidine hydroxylase (sequence ID: WP_051702360) from *Streptomyces vinaceous*. Moreover, microviridin precursor peptide sequence in the *Cyanocohniella* sp. LLY genome showed an 83% identity with the microviridin family tricyclic proteinase inhibitor gene of *Nostoc* sp. (sequence ID: WP_224090535.1). The capreomycidine and microviridin biosynthetic pathway included arginine hydrolase and condensation, adenylation, thiolation, epimerization, and reductase enzymes.

## 4. Discussion

Cyanobacterial biomasses are well-documented sources of metabolites for biotechnological applications and hopefully for humankind. Some cyanobacteria species are an inherited foodstuff from ancient cultures and are still consumed nowadays [1,2,6,7,10,16]. In South America, edible cyanobacterial macrocolonies known by the vernacular name of Llayta have been consumed since pre-Columbian times, and today they are available as dry biomass at food markets in northern Chile and southern Peru [10,16,19,20,21,22].

Andean wetlands over 4000 m of altitude are habitats under daily and seasonal environmental stressors, where resilient life forms have evolved morphological and biochemical adaptive mechanisms to withstand high solar UV insolation, desiccation periods, pH, osmotic and temperature changes, and other environmental stressors [44]. The outer envelope and the mucilaginous inner matrix of cyanobacterial macrocolonies are physical protective barriers against solar radiation and the loss of intracellular water [26]. Then, genomic information on functional capabilities of the Llayta microbiome may be understood as part of the adaptive responses of this microbiome to survive and proliferate under the stressful environmental conditions found at the Andean wetlands where these macrocolonies can be found.

*Nostocales* are morphologically diverse and ubiquitous to almost any ecosystem on Earth, including those under extreme environmental conditions. Trichomes containing vegetative and N_2_-fixing heterocyst cells develop during the life cycle of diazotrophic species. Cyanobacterial macrocolonies can be found in nature as sheet-like or spherical forms, with green to brown colorations, depending upon the species, dehydration state, local environment, with a massive accumulation of filaments in a mucilaginous matrix, rich in a complex polysaccharide, enclosed by an outer envelope [24,26,45]. In natural environments, cyanobacterial macrocolonies are colonized by diverse microorganisms involved in biogeochemical cycles and ecosystem services, although differences in heterotrophic bacteria diversity among the inner matrix, the envelope, or the surrounding environment of *Nostoc* macrocolonies have been reported [27,28,29]. Secker et al. [27] elucidated the absence of bacteria at the inner matrix of *Nostoc* macrocolonies from ephemeral wetlands in New Zealand, but high bacterial diversity at their external surface enriched with members of the genus *Sphingomonas*. Conversely, a different bacterial composition was found at the inner matrix, outer layer, and the littoral zone on *Nostoc* macrocolonies from Chungará Lake in northern Chile [28]. Comparatively, and using metagenomics, multiple taxonomic markers, and microscopic approaches, Satjarak et al. [29] reported high taxonomic diversity (cyanobacteria, microalgae, and anoxygenic bacterial genera) on the accompanying epimicrobiota of macroscopic dark-brown sheets of *Nostoc* from standing water pools at Lauca National Park in northern Chile. 

The accompanying microflora of edible Llayta macrocolonies has been poorly explored and its presence becomes evident during attempts to isolate axenic trichomes. Vilo et al. [45] reported the identification and the draft genome of a *Bacillus* bacterium from the microbiota associated with Llayta colonies, as a first approach to address physiological relationships within the Llayta microbiome and to inquire into the survival and adaptive strategies to arsenic, metals, UV radiation, and other prevalent extreme environmental stressors in the Andes wetlands. In this context, the present report provides new metagenomics-based information, as a culture-independent approach to improve our understanding of the Llayta cyanosphere and to gain insights into the metabolic capabilities of Llayta cyanosphere and natural products prospection. 

The metagenomics analyses on the cyanosphere of Llayta filaments provided proper sequencing data for identification and genome reconstruction of strain *Cyanocohniella* sp. LLY. The *Cyanocohniella* genus, Order Nostocales, is a recently described and revised genus with a filamentous diazotrophic holotype species *Cyanocohniella calida*, isolated from a thermal spring, Czech Republic [46]. The complex life cycle of *C. calida* shows similarity with *Pseudanabaena*-like stage, *Nostoc*-like stage, and *Chlorogloeopsis*-like stage, which makes the taxon especially difficult to identify from environmental samples. More recently, *Cyanocohniella crotaloides* sp. nov. was obtained after sampling sandy beach algal mats in Netherlands and described as showing a polymorphic life cycle resembling *Nostoc*-like thallus structures with curved, constricted filaments embedded in a nonlamellated, fine colorless sheath [47]. These authors also provide a detailed morphological comparison between these two *Cyanocohniella* species. In addition, strain TAU-MAC 3117 has been proposed as a third new species, *Cyanocohniella hyphalmyra*, isolated from epilithic mats on the surface of rocks in the shore of a brackish lake in Greece and found positive for HCN production [48]. Finally, two other isolates, SY-1-2-EE and SY-1-2-Y, from biological soil crusts in the surroundings of saline potash tailing piles, were assigned to the *Cyanocohniella* genus based on SSU rRNA and ITS sequences [49]. More recently, Jung et al. [50] have point out that all known members of the *Cyanocohniella* genus have been isolated from salty habitats and this genus might have had a marine origin and expanded to colonize salty terrestrial habitats. Based on our work, strain *Cyanocohniella* sp. LLY was identified as a new member of the *Cyanocohniella* genus and found in the cyanosphere of filaments of Llayta macrocolonies developing naturally in brackish water in Andean wetlands. 

Gene mining conducted on the reconstructed genome of *Cyanocohniella* sp. LLY demonstrated the presence of conserved lycopene cyclase *cruA/cruP* genes and complete *mysABCD* and *scyABCDEF* gene clusters involved in mycosporine-like amino acids and scytonemin biosynthesis, two UV-absorbing molecules located at the outer cell membrane and the intracellular compartment, respectively [51]. This genomic analysis shows putative synthetic capabilities of *Cyanocohniella* sp. LLY needed for protection against photooxidative damages triggered by high-UV solar insolation prevailing at the Andes wetlands [14,36].

Lasso peptides are ribosomally synthesized and post-translationally modified small peptides with antimicrobial, antiviral, and enzymatic inhibitors [52]. The gene cluster for Lasso peptide biosynthesis includes the precursor peptide, an ATP-dependent cysteine protease, and one ATP-dependent macrolactam synthetase. The general cluster organization has been studied in Actinobacteria, Firmicutes, and Proteobacteria phyla showing differences in the presence of an ABC-transporter gene and a splitting of the B proteins in Actinobacteria in contrast to the Proteobacteria cluster. However, both gene organizations can produce the Lasso peptide [52,53,54]. Gene mining on the genome of *Cyanocohniella* sp. LLY showed the presence of all genes needed for lasso peptide biosynthesis with a similar organization found in Actinobacteria [54]. Although the gene cluster in *Cyanocohniella* sp. LLY has a reverse direction for transcription (Figure 3A) compared to the Actinobacteria cluster, it might produce the necessary peptides for the biosynthesis of Lasso peptide. According to phylogeny, the precursor peptide sequence is conserved in Nostocales (Figure 3B) and would confirm the genomic potential of strain LLY for Lasso peptides biosynthesis.

Gene mining of the assembled genome of *Cyanocohniella* sp. LLY showed the potential capability for capreomycidine biosynthesis and the production of members of the tuberactinomycin family of antibiotic peptides, such as viomycin and capreomycin [55], but also for the biosynthesis of a microviridin precursor that would render a serine-protease inhibitor peptide [56]. 

The bioinformatic analysis of the metagenome-assembled genome from *Cyanocohniella* sp. LLY showed the presence of CRISPR-Cas operons conferring an adaptive immunity defense to the Andean cyanobacterium against mobile genetic elements such as phages. Such defensive systems have been reported in archaea and bacteria [39,57], organized as CRISPR arrays (short, direct repeats of 20–40 bp separated by variable spacers) and Cas operons [39,40]. According to Shao et al. [42], the Cmr complex binding and cleaving of foreign RNA may involve crRNA endoribonuclease activity coded at one of the *Cmr1*, *Cmr3*, *Cmr4*, or *Cmr6* genes. In contrast, the *Cmr2* gene is involved in binding specific targets. The crRNA transcribed by the CRISP array would bind to CAS proteins to recognize the foreign nucleic acid, which would be cleaved by Cas1 and Cas2 nucleases [37,38,42]. Sequences for CRISPR-CAS system operons found at the metagenome-assembled genome from *Cyanocohniella* sp. LLY were substantially similar sequences from CRISPR-CAS operons available in public databases.

## 5. Conclusions

Metagenomic analyses of the cyanosphere of filaments isolated from edible Llayta macrocolonies allowed us to identify the presence of the cyanobacterium strain *Cyanocohniella* sp. LLY. Its reconstructed genome provided information regarding its putative capabilities for the biosynthesis of UV-absorbing molecules (mycosporine-like amino acids and scytonemin), Lasso peptides, microviridins, and the presence of CRISPR-CAS system operons. The assembled genome of strain LLY is now available for further inquiries and may be instrumental for taxonomic advances on this genus as well as concerning its origin and spatio-temporal distribution [50,58]. All known members of the *Cyanocohniella* genus have only been isolated from salty European habitats. The existence of strain *Cyanocohniella* sp. LLY comes to fill a biogeographic gap, including the Andes Mountains wetlands in South America as a new habitat for the genus.

The Atacama Desert is a biodiversity hotspot deserving extensive research and environmental protection. Edible Llayta macrocolonies from high-altitude Andean wetlands are another example of its microbial richness [59]. Further studies relating to the cyanosphere associated with Llayta will render insights regarding the microbial functional capabilities, biosynthetic pathways, and adaptive strategies to the environmental conditions in high-altitude Andean wetlands.

## Figures and Tables

**Figure 1 microorganisms-10-01517-f001:**
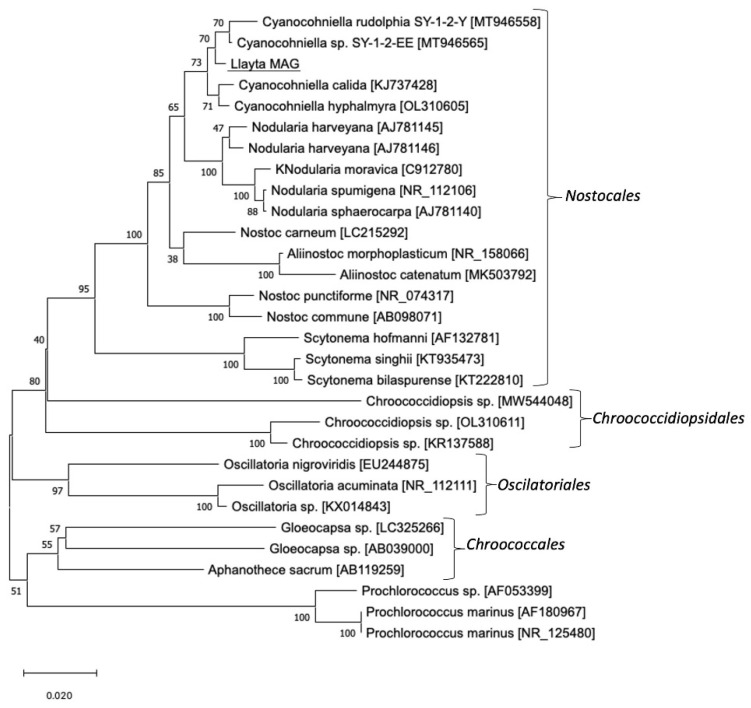
Phylogenetic analysis by Neighbor Joining method based on gene sequences of *16S rRNA*. The horizontal bar at the figure’s base represents 0.02 substitutions per nucleotide site. The percentage of trees in which the associated taxa clustered together is shown next to the branches, using a bootstrap of 1000. Evolutionary analysis was conducted in MEGA X.

**Figure 2 microorganisms-10-01517-f002:**
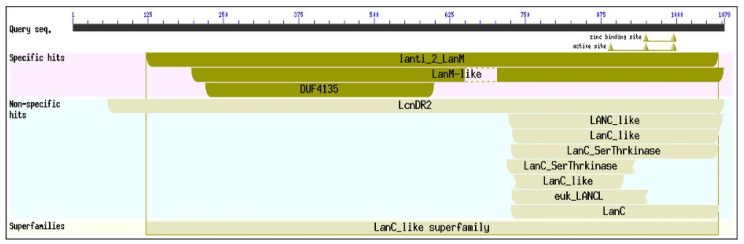
Domain characterization of lantibiotics biosynthesis enzyme using the NCBI Conserved Domain Database.

**Figure 3 microorganisms-10-01517-f003:**
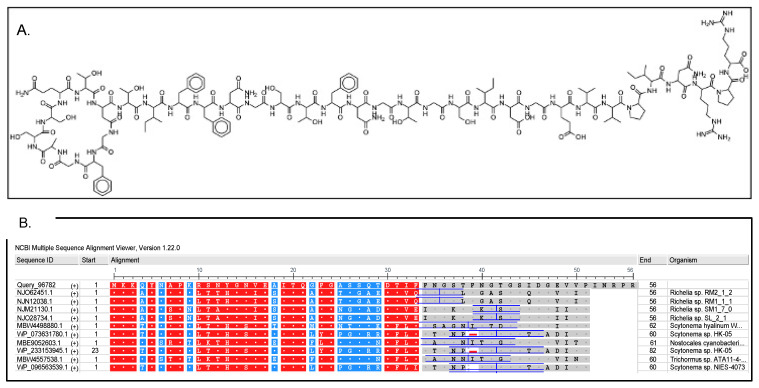
Characterization of the lasso peptides from the reconstructed genome of the *Cyanocohniella* sp. LLY cyanobacterium. (**A**) Primary chemical structure of the cleaved Lasso P, made of 36 residues. (**B**) Protein sequence alignment of Lasso P peptide against NCBI non-redundant database.

**Table 1 microorganisms-10-01517-t001:** *Cyanocohniella* sp. LLY open reading frames annotations and best Blast hit for *MysABC* and *ScyABCDEF* cluster genes.

ORF	Description Based on Subsystem Annotations	Accession	Identity (%)
Orf 3147	Demethyl 4-deoxygadusol synthase MysA	WP_069074324.1	93
Orf 3148	O-methyltransferase MysB	BBC27542.1	85
Orf 3149	ATP-grasp ligase forming mycosporine-glycine, MysC	BBC27543.1	82
Orf 4592	Scytonemin biosynthesis protein ScyA	WP_206262883.1	83
Orf 4593	Tryptophan dehydrogenase ScyB	WP_086764771.1	84

**Table 2 microorganisms-10-01517-t002:** *Cyanocohniella* sp. LLY open reading frame annotations and best Blast hit for CRISPR-CAS system operons.

ORF	Description Based on Subsystem Annotations	Accession	Identity (%)
Orf 2275	CRISPR-associated RAMP Cmr2	WP_179048547.1	92
Orf 2276	CRISPR-associated RAMP Cmr3	WP_102220820.1	95
Orf 2277	CRISPR-associated RAMP Cmr4	WP_179048549.1	94
Orf 2278	CRISPR-associated RAMP Cmr5	WP_102220822.1	95
Orf 2279	CRISPR-associated RAMP Cmr6	WP_179048551.1	78
Orf 2288	CRISPR-associated protein Cas1	WP_218653184.1	94
Orf 2289	CRISPR-associated protein Cas2	WP_179048556.1	94
Orf 3613	CRISPR-associated negative autoregulator Cas7/Cst2	WP_194144297.1	94
Orf 3614	CRISPR-associated protein Cas5	MBW4428053.1	96
Orf 5125	CRISPR-associated protein Cas2	WP_190898341.1	95
Orf 5126	CRISPR-associated protein Cas1	MBN3899475.1	93
Orf 5227	CRISPR-associated RecB family exonuclease Cas4	MBW4675378.1	91
Orf 5128	CRISPR-associated endoribonuclease Cas6	WP_096682797.1	94
Orf 5137	CRISPR-associated helicase Cas3	WP_179075640.1	97
Orf 5452	CRISPR-associated RAMP Cmr2	MBD2365142.1	92
Orf 5453	CRISPR-associated RAMP Cmr3	WP_190709980.1	95
Orf 5454	CRISPR-associated RAMP Cmr4	WP_190709978.1	93
Orf 5455	CRISPR-associated RAMP Cmr5	WP_190709976.1	95
Orf 5456	CRISPR-associated RAMP Cmr6	WP_190709975.1	91

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
