# Peer review of "Metagenome-Assembled Genome of Cyanocohniella sp. LLY from the Cyanosphere of Llayta, an Edible Andean Cyanobacterial Macrocolony"

_microorganisms, 2022, doi:10.3390/microorganisms10081517_

Round 1

Reviewer 1 Report

The revised manuscript presents new microbiome of Llayta cyanobacteria and some its metabolic abilities. In addition, these organisms are related to quite harsh environment (desert). Such type of studies are very important and required providing new knowledge about potentially valuable organisms.

For that reason I found the manuscript valuable and falling into the scope of Microorganisms Journal.

It has informative title, good abstract and keywords.

Introduction is quite short just describing the area of sampling, the role of Llayta as natural food resource and the need of studying the microbiome what was presented as the aim of the study.

Methods are clear and properly used, data were processed and deposited in GenBank. The results are sound presenting both taxonomic and functional analyses included. These analyses provided information about target organisms.

Results are discussed in the frame of similar microorganisms.

For me this manuscript is good for publication.

Author Response

Thank you for your comments on layout and content of our manuscript; we also appreciate your decision on the acceptance of our work for publication.

Reviewer 2 Report

Comments to the Authors

Vilo et al.

Metagenome-assembled genome of Cyanocohniella sp. LLY from the cyanosphere of Llayta, an edible Andean cyanobacterial macrocolony

The authors decided to substantially streamline the manuscript and to delete misleading figures, analyses and conclusions. I acknowledge that wrong interpretations about associated heterotrophic bacteria of Llayta macrocolonies were removed and that the revision was focused on the cyanobacterial MAG, even if the phylogenetic positioning of Llayta in the cyanobacterial tree of life is irritating because it is just based on the simple 16S-rRNA gene analysis. However, the authors considered my taxonomic criticism, the establishment of the Llayta MAG is sound and the relevance of this ancient cyanobacterial food resource justifies a publication. The metagenome paves the way for genome-based comparison with novel cyanobacterial ‘life style’ food products.

Points of criticism and suggestions

--- Abstract: Lines 17-19 and 24-25

The two sentences are nearly identical. Please remove one of them and modify the abstract.

--- Lines 80-81

No changes or explanations (See first review).

--- Lines 93-43

The phrase “resembling the four most abundant bacterial genera” is not meaningful. As a microbiologist, I have no idea which bacterial genera are the most abundant ones…

Please clarify this sentence e.g. by “resembling the four abundant bacterial genera [‘genus name 1’] (Alphaproteobacteria)], [‘genus name 2’](Bacteroidetes), [‘genus name 3’]…”

--- Line 95-65

It should say: “Most importantly, the metagenome analysis revealed that 16% of the reads corresponded to the phylum Cyanobacteria, from which one cyanobacterium genome was reconstructed by metagenomic binning. The MAG was assigned to the genus Cyanocohniella based on 16S-rRNA gene analysis, which is…”.

--- Line 101

The binning strategy…”.

--- Line 104

“…belonged to the order Nostocales”.

--- Line 104

Bootstrap support of 56% and 79% is not absolutely “clear” and the phrasing is not sound. It should say “Moreover, the 16S rRNA gene analysis revealed a placement of Llayta within a clade of Cyanocohniella sequences (Fig. 1). A BLASTN search with the Llayta MAG’s 16S rRNA in the NCBI nr database showed a sequence identity of 99.2% with Cyanocohniella sp. SY-1-2-Y (NCBI accession number: MT946558.1).

--- Page 3, Figure 1

(1)   The 16S rDNA sequence of the closest relative “Cyanocohniella sp. SY-1-2-Y” should be included in the simple 16S rRNA gene tree.

(2)   The accession numbers of the reference sequences should been presented in brackets behind the taxon name.

(3)   The 16S rDNA of Llayta should be highlighted in bold.

(4)   The affiliation of different taxa to the orders Nostocales, Chroocccidiopsidales, etc. should indicated by a vertical line behind the taxon names.

--- Lines 146-150

The authors should correctly distinguish between protein names (upper case, no italics [LanM]) and gene names (lower case, italics [lanM]) throughout the manuscript. Based on the protein accession number HAX78725.1 it should say “… contained sequences of the protein LanM, involved…”.

Author Response

We thank Reviewer for the constructive criticisms, comments and suggestions on our manuscript. We have edited our manuscript following all your comments.

Abstract: Lines 17-19 and 24-25: Abstract has been removed and modified according to the Reviewer comment.

Lines 80-81: the section mat and methods and results has been edited according to the Reviewer comment

Lines 93-43: the sentence has been modified and genus names were included.

Line 95-65: the paragraph has been edited according to the Reviewer comment.

Line 101: it was edited according to the Reviewer comment.

Line 104: it was edited according to both Reviewer comments.

Page 3, Figure 1: all three corrections have been included on Fig. 3.

Lines 146-150: names of proteins and genes have been edited.

Reviewer 3 Report

Metagenome-assembled genome of Cyanocohniella sp. LLY from the cyanosphere of Llayta, an edible Andean cyanobacterial macrocolony by Claudia et al., 

1. Line 30-34: "Cyanobacterial and microalgal biomasses have been consumed for centuries and Dunaliella, Chlorella and Arthrospira (in Africa and North America), and Nostoc (in Asia and South America) are documented sources of essential amino acids, vitamins, polyunsaturated fatty acids, carotenoids, phycobiliproteins, and secondary metabolites with diverse biotechnological applications" 

Provide reference separately, for example "....are documented sources of essential amino acids (ref), vitamins (ref), polyunsaturated fatty acids (ref), carotenoids (ref), phycobiliproteins (ref), and secondary metabolites (ref)...."

2.  Isolation Llayta filaments- should be changed to "Isolation of Llayta filaments".

3. Provide versions used for each online platforms/softwares used in section 2.

4. Use } on figure 1 to mark the excat separation for each orders. 

5. Line 117-132: It is not clear that how the functional annotation for the genomes were performed.

6. Line 126: "Genomic analysis in silico demonstrated..." - correct the sentence.

7. How table 1 was generated, was it generated from Antismash and then blasted.

And similarly, for all the secondary metabolites detected, its not clear that if it was generated from Antismash or some other software used.

As stated in methodology that PRISM and Antismash both were used, but i dont see a clear presentation in the result section about these platforms.

More importantly, i don't see any novelty in the work except for reporting the genomic data.

Author Response

We thank and appreciate the reviewer extensive revision and for the constructive criticisms, comments, and suggestions on our manuscript. We have edited our manuscript following all your comments.

  1. Lines 30-34: References have been reorganized as requested and Reference List has been modified accordingly.
  2. The change has been made.
  3. Versions of online platforms/software used in section 2 have been provided.
  4. Fig. 1 has been edited accordingly.
  5. Line 117-132: the text was edited to include how functional annotation of genomes was performed.
  6. Line 126: The sentence has been corrected.
  7. Table 1. and results section have been modified and improved following the reviewer thoughtful comments

Round 2

Reviewer 3 Report

The manuscript has been improved substantially except for the novelty of the work which I see more as reporting genomic data. 

I only have one minor point, which needs to be addressed before accepting it for publication:

Line 90-93: "Sequencing reads are available at the Sequence Read Archive (SRA) with accession number SRR17916224. The Whole GenomeShotgun project have been deposited at DDBJ/ENA/GenBank under the accession JAKOMP000000000."

Should be presented as a separate section -Data Deposition:

"Sequencing reads are available at the Sequence Read Archive (SRA) with accession number SRR17916224. The Whole Genome Shotgun project has been deposited at DDBJ/ENA/GenBank under the accession JAKOMP000000000."

This manuscript is a resubmission of an earlier submission. The following is a list of the peer review reports and author responses from that submission.

Round 1

Reviewer 1 Report

Comments to the Authors

Vilo et al.

Genomic insights on the functional capabilities of the cyanosphere of edible Nostoc macrocolonies (Llayta)

The authors provide a metagenome study from edible macrocolonies of cyanobacteria initially obtained from a Chilean farmers market. The dried filaments were rehydrated, grown in nitrogen-free medium and treated with gentamicin. Isolated DNA from the gentamicin-resilient cyanobacterial enrichment culture was Illumina-sequenced, assembled and finally binned. Metagenome binning resulted in six MAGs including the cyanobacterium and five associated Proteobacteria (4x alpha, 1x gamma) of the cyanosphere. The authors tried to deduce the microbial diversity from the metagenome data and performed phylogenetic analyses to determine the closest relatives of the cyanobacterium (Llayta) and the associated heterotrophic bacteria. Furthermore, they searched for secondary metabolite gene clusters within the metagenome.

The background of neglected natural cyanobacterial food resources from the pre-Columbian times in South America is fascinating and I was really curious about the results of the current study. However, there was a great discrepancy between the narrative of the well written introduction/discussion and the reliability of the bioinformatic analyses resulting in misleading conclusions. The establishment of the MAGs including the deposition at the NCBI database seems to be sound, but I have four major points of criticism regarding the rather poor quality, reliability and interpretation of the presented in silico analyses.

Major points of criticism

(01) Abundance of taxa on genus level

--- Lines 114-129; Figure 1

My first major point of criticism is the completely misleading outcome of Figure 1. Data analysis is not adequately described in the methods section, but I suppose that the authors simply analyzed all assembled contigs of the metagenome via BLASTN searches and presented the respective outcome. This proceeding is usually used for 16S-rRNA gene amplicon analyses and provides – based on a single comparable marker gene – reliable insights into the bacterial diversity of a sample. However, a comparable analysis with the established metagenome data is not meaningful as clearly documented by Figure 1, which is predicting that cyanobacteria of ten different genera are present in the low complexity Llayta enrichment culture. Already based on their own data this conclusion is absolutely wrong! (i) Metagenomic binning just resulted in a single cyanobacterial MAG. (ii) Analyses with draft genomes of axenic cyanobacteria would provide comparable results, i.e. a bioinformatic artifact proposing cyanobacterial pseudo-diversity. This artifact can be simply explained by lack of a closely related reference genome thus resulting in contradictory best BLAST-hits of contigs belonging to one MAG. Accordingly, at least the “Nostoc”, “Anabaena” and “Nodularia” hits that are presumably representing 87% of the cyanosphere can be traced back to the Llayta MAG. The same misleading data are presented in the left part of Figure 1, where the ten “detected” genera probalby correspond to the five assembled proteobacterial genomes.

It is still possible that further non-dominant heterotrophs or even additional cyanobacteria are present in the enrichment culture of Llayta, but this prediction could be tested by an exhaustive search for 16S-rRNA genes in the metagenome. Furthermore, the authors should provide information about additional bins with a completeness of less than 91.3% as supplemental table. Table 1 & Table S1 should also contain the CheckM-based contamination level of the respective MAGs. Additional coverage information of the different MAGs would allow to draw conclusions about the abundance of the cyanobacterium and the five associated heterotrophs within the cyanosphere.

Recommendation: Figure 1 and the corresponding results should be completely removed from the manuscript. A meaningful phylogenetic analysis of the five proteobacterial MAGs could be presented instead.

(02) Phylogenetic analysis of the Llayta MAG

--- Lines 162-180; Figure 3

The phylogenetic analysis and corresponding conclusion about the taxonomic assessment of the Llayta MAG as Nostoc sp. is also completely misleading. It essentially resulted from a scientific circular argument based on the probably arbitrary cyanobacterial taxon sampling that was restricted to seven Nostoc reference genomes. An analogous taxon sampling of seven Synechococcus genomes, which also represents a paraphyletic genus, would result in an comparable phylogeny, where the Llayta MAG is located among Synechococcus strains with low statistical support! 

A meaningful phylogenetic analysis would require a taxon sampling of at least 30 reference genomes that are chosen based on their close phylogenetic relationship! These genomes can be identified by simple 16S-rDNA BLASTN searches or by BLASTP searches with any unique marker protein.

I could identify a 16S-rDNA gene (locus tag: MET_12195) on the 55,410 bp contig JAKOMP010000029.1 from the Llayta MAG. A BLASTN search of the complete contig confirmed its authentic cyanobacterial affiliation. Furthermore, a BLASTN search with the 16S-rRNA gene against the type strain database showed a close affiliation with Cyanocohniella crotaloides (98.29% identity; MN243143.1) and Cyanocohniella hyphalmyra (98.67% identity; OL310605.1), which clearly documents that the Llayta MAG belongs to the genus Cyanocohniella! Furthermore, BLASTN searches in the non-redundant NCBI database revealed Cyanocohniella sp. SY-1-2-Y with 99.19% 16S-rRNA gene identity as closest relative (MT946558.1). Images of this isolate can be found in Figure 18a,b of the paper from Sommer et al. (2020) published in the journal Microorganisms (doi:10.3390/microorganisms8111667).

A simple, but well-performed 16S-rRNA gene tree with a meaningful taxon sampling would be sufficient for the correct taxonomic classification of the Llayta MAG. If the authors prefer to present a phylogenetic analysis performed on concatenated genes, I strongly recommend a tree based on unique protein sequences. The usage of Bacillus pumilus as an outgroup sequence make no sense, three or four distantly related Synechocystis strains could be used as a suitable outgroup.

Recommendation: It is a must that the misleading Figure 3 has to be replaced by a meaningful phylogeny that represents the true phylogenetic affiliation of the established Llayta MAG.

(03) Autheticity of putative Llayta genes/gene clusters

--- Lines 181-264; Figures 4 & 5

A crucial point in metagenomics is the occurrence of non-authentic sequences in a MAG due to binning problems. Accordingly, I recommended above to present the contamination rate in Table 1, because this value allows an estimation of the quality of the investigated MAG. A (bioinformatic) contamination level of below 3% is a valuable indicator for a barely contaminated bin.

However, the authors should validate that their genes and gene clusters that are presented in chapter “3.4. Functional capabilities of Cyanocohniella sp. Llayta” are authentic cyanobacterial genes. The check could be simply based on a BLASTN search of the complete contigs, where the genes are located on, against the NCBI nr database (analogous to my quick investigation of the 16S-rDNA contig). This check is especially important for the putative lasso peptide gene cluster of Llayta that is homologous to those of Actinobacteria. If the cyanobacterial affiliation is validated, this information should be mentioned in the manuscript, if the BLAST-searches revealed a non-cyanobacterial affiliation the respective passage should be removed from the manuscript.

(04) Content and length of chapters (results)

--- e.g. lines 149-158; Chapter 3.2

A paragraph in the results section comprising of a single sentence is not meaningful! The MG-RAST subsystem annotation revealed many categories of genes that have nothing to do with the adaptation to extreme environments and genes for fatty acid or DNA metabolism would also be found in any E. coli strain. If the authors want to draw this conclusion based on the presented genome data, they should contrast the outcome in comparison to another filamentous cyanobacterium that has been isolated from a mesophilic habitat. Otherwise, the conclusion would be again a circular argument that just represents a (plausible) working hypothesis.

General recommendation: After fundamental revision of the results section, the abstract and conclusion have to be modified according to the corrected outcome of the analyses.

Minor points of criticism (non exhaustive)

--- Figures 1

Many figures have a poor quality and seemed to be simply copied from a screenshot. Please prepare them in an adequate form for the publication.

--- Line 78

The authors should show or cite the recipe of the “Allen and Arnon” (?) medium.

--- Line 83

The rationale for the usage of gentamicin instead of any other antibiotic is unclear.

--- Lines 86-88

What was the reason for adding additional carbon sources that enhances the growth of heterotrophic proteobacteria?

--- Line 97-99

If the average insert size of the library is 250 bp, the corresponding read length can´t be 300 bp! Please clarify.

--- Lines 157-158

Please refer to the software that was used to create the Krona plot.

--- Line 211

It should say “the protein LanM”.

--- Lines 236-237; Figure 5A & Lines 354-355

The precise inversion of all four genes in the upper or lower gene cluster is not plausible! Please carefully check the structural composition.

--- Lines 257-258

Incomplete sentence; wording.

Reviewer 2 Report

The revised manuscripts presents interesting insight into the microbiota associated with filaments of Nostoc sp.

This topic is interesting as this is a natural food present on local markets. Additional metagenomic and metabolomic analyses will give more information on this food-product. In addition, new knowledge on any microbiomes and their possible application is always interesting and useful.

Here, Authors collected well known in their region "product" and used metagenomic tools for assessing its structure and function. This was the right procedure providing reliable results.

Taking into account all of above I found the manuscript interesting, done well and falling into the scope of Microorganisms MDPI Journal.

There are, however, few things needing improvement

  1. I would recommend language improvement as there are many grammar mistakes (the list is below)
  2. The figures quality is very low (especially font) making them hard to read

Specific comments:

L17: Correct 'its' into 'their'
L19-20: please write rather "however, the cyanoshpere of Llayta microcolonies...." than "however, the Llayta microcolonies cyanosphere...." 
L24: use 'bacterial genomes' not 'bacteria genomes'
L25: I think here should be 'gene clusters'
L26, 72: The word 'insight' is followed by rather 'into' than 'on'
L33: Latin names should be in italics (Dunaliella, Chlorella and Arthrospira.)
L40: 'liquid water' - I understand the intention, but word water is rather used for liquid state (gas is vapour or steam depending on temperature). I would suggest to use simply 'water'; alternatively "water in a liquid state"
L49: I think when giving altitude value it'd be more better to use "at altitude of over 3,000 m" or alternatively "3,000 m above sea level"
L77: Please add a preposition 'a' - "in a sterile.."
L79: upper-indices in units are missing
L90: Please add a preposition 'a' - "in a liquid medium"
L97: Please remove '_' between two sentences -  (._The)
L111: Please add sp. after Nostoc 

Isolation procedure
I'd like to ask to add a little bit information about samples material. I wonder how many samples were bought (one seller or more) and what was the amount of each sample. I'd would be good to know if there were sufficient amount of replications for all analyses.

The second issue, why no to collect fresh biomass? I wonder about microbiome differences due to drying. Are there any data about this issue? Maybe this is a topic for new study - direct sampling in the field.

Other parts of study are good for me and can be trusted.

Reviewer 3 Report

The article " Genomic insights on the functional capabilities of the cyanosphere of edible Andean Nostoc macrocolonies (Llayta)" by Claudia et al.,

The manuscript describes the metagenomic analysis of edible Andean Nostoc macrocolonies (Llayta) with a detailed description of their metabolic potential. 

I personally feel that just the metagenomic analysis is not enough for publication in this journal.

Apart from the limited data presentation, I also have some doubts regarding the methodology used by the authors:

  1. Why the colonies were treated with Gentamycin, is there any specific reason behind it or was just used to avoid any other lab based contaminatto be sequencesd. In any case gentamycin can also inhibit some of the bacteria present in the cyanosphere. 
  2. The secod most abundant cyanobacteria was belonging to the genus Anabaena followed by Nodularia. But no detailed description was provided further in the manuscript regarding these genera.
  3.  What version of Antismash was used for analysis of BGCs. Detailed results for Antismash should be presented in better form.
  4. Abstract states that the current work provides insights into the ecological role and adaptive strategy? how?
  5. What about the genome analysis for Paracoccus, Microvirga, Mesorhizobium, Blastomonas and Aquimonas. It will be good to understand the secondary metabolites from these bacteria too as they are the part of the edible Llaya.

References should be checked properly...For example Line 271, Authors cited ref no. 8, 12 and 30. Whereas ref 30 is about algae as food not cyanobacteria...